# A Study on the Factors Influencing the Sustainable Development of Education in the Context of COVID-19: Tencent Conference Online Platform

**Xin Wang [1], Xingmeng Ma [2], Ziyi Wang [1]** and **Yanlong Guo [2],***

1   Anhui Cultural Tourism Innovative Development Research Institute, Anhui Jianzhu University, Hefei 230601, China; wangxin020015@ahjzu.edu.cn (X.W.); ziisevenyyoo@gmail.com (Z.W.)
2   Social Innovation Design Research Center, Anhui University, Hefei 203106, China; n22301048@stu.ahu.edu.cn
*   Correspondence: 20106@ahu.edu.cn; Tel.: +86-152-5655-6306

**Abstract:** The difficulties of offline education caused by the COVID-19 epidemic are attracting increasing public attention. Although international research on online education platforms is gradually deepening, the research on the influencing factors of Chinese users' participation in these platforms shows different results due to regional differences. Therefore, this study adopts the technology acceptance model (TAM) as the basis to build the index system of influencing factors for the Tencent conference online education platform. The questionnaire design covered five dimensions, including external environment, platform satisfaction, and continuous use intention. An online questionnaire survey was conducted on participants in some regions of China, 320 questionnaires were distributed, and 297 valid questionnaires were collected. The data were analyzed by SPSS 26.0, and the results of Cronbach's coefficient of 0.939 and a KMO value of 0.941 were obtained. The weight value, entropy value, and importance ranking of the factors were determined by combining hierarchical analysis and the entropy weighting method. First, the perceived ease of use is the most critical factor influencing the users' use of online delivery platforms. Second, freedom from geographical time difference, the ease of operation, system perfection, and proficiency in use greatly influenced the participants' use of the online lecture platform. Third, the study group suggested that the platform should be optimized in terms of convenience, stability, and freedom from geographical time difference, and provide reasonable references and lessons for future educational efforts.

**Keywords:** web-based delivery; Tencent conference; technology acceptance model; hierarchical analysis; entropy power method

## 1. Introduction

In the winter of 2019, cases of the novel coronavirus were reported in Wuhan, Hubei Province, China [1]. At the beginning of the following year, the World Health Organization named the outbreak coronavirus disease (COVID-19) [2,3]. The outbreak is now having a profound impact on the health, work, and education of people around the world. The United Nations 2030 Agenda for Sustainable Development is an international initiative to address increasingly pressing global challenges and promote sustainable development worldwide. Ensuring inclusive and equitable quality education and lifelong learning opportunities for all is the fourth goal of this agenda. According to the requirements of the Ministry of Education of the State, enterprises and institutions have established platforms such as "cloud classroom", "classroom in the air", and "online classroom", which to a certain extent solves the problem of the lack of online classrooms in schools [4,5]. In recent years, with the development of the Internet and new media technology, online office platforms have flourished [6]. Online platforms such as Tencent conference (Shenzhen, China) and Massive Open Online Course (MOOC) (Guangzhou, China) are widely used.

Online education platforms independently developed by Chinese universities or jointly initiated by social entities mainly include China University MOOC (Guangzhou, China), Xuetang Online (Beijing, China), Good University Online (Shanghai, China), and Superior Course Alliance (Shanxi, China). The representative online education platforms developed and operated by Internet enterprises mainly include the following: NetEase Cloud Class, Tencent Class, New Oriental Online, etc. At the same time, educational platforms in other countries have their own unique advantages. For example, platforms such as Khan Academy and Coursera in the United States are favored by users around the world for their rich course resources and high-quality teaching content. FutureLearn in the UK has attracted many users with its flexible learning methods and diverse course content. In contrast, China's education platforms have significant advantages in terms of content richness, teaching methods, and user experience.

Among many online teaching tools, China has developed video conferencing software (3.24.3) [7]. Tencent conference is an online learning system with the feature of simulating the face-to-face teaching of users in a real environment [8]. Compared with other online teaching platforms, Tencent conferencing exerts the advantages of stability and simulation, aiming to improve teaching efficiency [9–12]. Tencent conferences feature high-definition video and audio calling capabilities, allowing users to interact almost without latency. Its flexible screen-sharing feature enables teachers to present teaching content in real-time. At the same time, with instant messaging and group discussion room functions, students can easily have group discussions or communicate one-on-one with teachers.

Against the background of distance education playing an increasingly important role in daily teaching and learning activities, it is particularly important to study what factors influence people's perceptions of using online teaching and learning platforms. The origins of the idea of the value of education for sustainable development can be traced back to the 1987 UN General Assembly document "Our Common Future". The formation of sustainable development values has provided the content and practical basis for value education, and further enriched the theoretical connotation of the concept of ESD values.

The overall aim of this study was to explore the key factors affecting the sustainable development of education in the context of the COVID-19 pandemic, with a particular focus on the use of online conferencing software. Tencent Conference (3.24.3) was chosen as the research object because it was widely adopted as a major tool for teaching and business communication during the pandemic. The team's findings underscore the importance of interdisciplinary collaboration in developing and supporting online learning platforms, which echoes the Special Issue's emphasis on interdisciplinary education. In addition, the findings provide valuable insights for education policymakers and curriculum designers who can use them to improve learning outcomes when designing and implementing sustainable distance education solutions.

Indeed, if users are found to have a positive attitude towards online conferencing software, this indicates that the software is more likely to be adopted by university faculty and business professionals. This is especially important for higher education institutions, as it involves curriculum design, innovation in teaching methods, and the effective use of distance education resources. This research not only provides a theoretical and empirical basis for understanding and promoting the sustainable use of online education platforms but also provides practical recommendations for the sustainable development of higher education, thereby directly supporting the core objectives and scope of this Special Issue.

## 2. Literature Review

To date, academics across the globe have highlighted the importance of online teaching in terms of educational sustainability. According to Wea, N and Kuki, A, the unique format of online courses breaks the traditional concept of teaching and learning. Online teaching broadens the scope and methods of education and learning [13]. Based on digital technology, educators in Europe are upgrading their digital skills through online teaching models [14]. Mäkelä et al. identified breaking down constraints, sharing resources, and timely feedback

as the main influences on online teaching and learning [15]. Some scholars have compared the effectiveness, advantages, and disadvantages of online instruction with traditional classroom instruction. They believe that online teaching should make changes in terms of technological limitations, disruptions, and instructor competence [16]. In Collaborative Learning (CL), interactive computers are used to teach literacy to deaf children. This is a good example of the strengths of Collaborative Learning in terms of uniqueness and relevance.

Positive progress is being made in integrating the concept of sustainable development into high-quality education and online teaching in schools. At the same time, network lecture is also affected by teaching space, time, content, and method. This is reflected in the fact that students often need to interact with multiple teachers and the diversity of online platforms that the teachers choose. In an epidemiologic context, issues and challenges in platform fluency, instructor–student interaction, and student autonomy were identified by Coman et al. [17,18]. Online teaching is a new form of communication and interactive platform. It provides information to users through the Internet [19]. Lu, H. et al. suggest that online teaching is an interaction that relies on the online resources of the Internet to realize the interoperability of information and competence [20].

According to the different research directions of online delivery platforms, scholars use different models and methods to analyze the relevant issues [21]. Within the scientific attitude and intention, the technology acceptance model (TAM) is considered the most widely used and influential framework [20]. Moreover, TAM is a complete model for finding theoretical frameworks in an academic context [22]. Studies that combine perceptual journeys with TAM demonstrate the cost of the human use of science [23]. It means to focus on a wide range of experiences as well as timely experiences, and strongly predicts the perceived usefulness and ease of use. Click data from online learning environments were analyzed by two mathematical methods by An-tonenko et al. to determine the learning behavior characteristics of high-achieving students [24]. Some scholars used a potential correspondence model of machine learning to detect whether students misuse intelligent tutoring systems for the sake of creating a classification system for detecting students' honesty rates [25]. Others combined the concept of self-selection and argued that the factors influencing participants' willingness to continue learning online are the basic indicators of TAM and perceived enjoyment [26].

The importance of online technological development is illustrated by the results achieved by scholars in terms of technological advances and tool use based on the Internet. The concept and actions of ESD have been further strengthened globally. Online delivery has become an increasingly popular educational tool [27]. Other scholars have argued that the continued use of online instruction can help improve integration with traditional instruction [28]. The rationalization and differentiation of school policies are critical to the use of Internet technology in teaching and research [29]. Linnes, C et al. found that online instruction is applicable to educational systems that incorporate the Internet, technology, and tools [30]. Another scholar argued that improvements in technology and the development of the Internet have been enhanced with the turn of the century. It has enhanced the dynamics of online learning [31]. However, Joshi et al. came up with a different result by stating that the success of online learning in terms of pedagogy is controversial as it leaves a lack of communication and opportunities between users [32]. Influenced by the COVID-19 pandemic and the international political and economic environment, global student mobility has shown a new trend. However, most people have adapted to online or blended learning and teaching [33].

In summary, scholars have analyzed the development of online lecture platforms from various aspects. They pointed out the advantages and disadvantages of online lecture platforms. However, the real value of ESD is manifested in the fulfillment of the educational needs of the educated for sustainable development. However, only the needs of the educated cannot produce real value, and there are also constraints in the knowledge, ability, interest, and will of the educated. Educational resources are the necessary conditions for

the sustainable development of education, and the sustainable development of education must rely on educational resources.

## 3. Research Methodology

In this section, we focus on the research methodology and questionnaire design to establish the basis for the subsequent processing and quantitative study of the indicators. First, the research team summarized and generalized the TAM model of perceived usefulness, the perceived ease of use, external environment, platform satisfaction, and willingness to continue using the platform. The model was constructed by studying the usage perceptions of Chinese participants to discover which factors affect users' willingness to use the online lecture platform. Secondly, the analytical hierarchy process and entropy method are used as the basis to explore relevant scholarly research theories. The purpose is to promote the establishment of online lectures and effective teaching. Finally, the process underlying the questionnaire design is elaborated. Specific methods and processes are shown below (Figure 1).

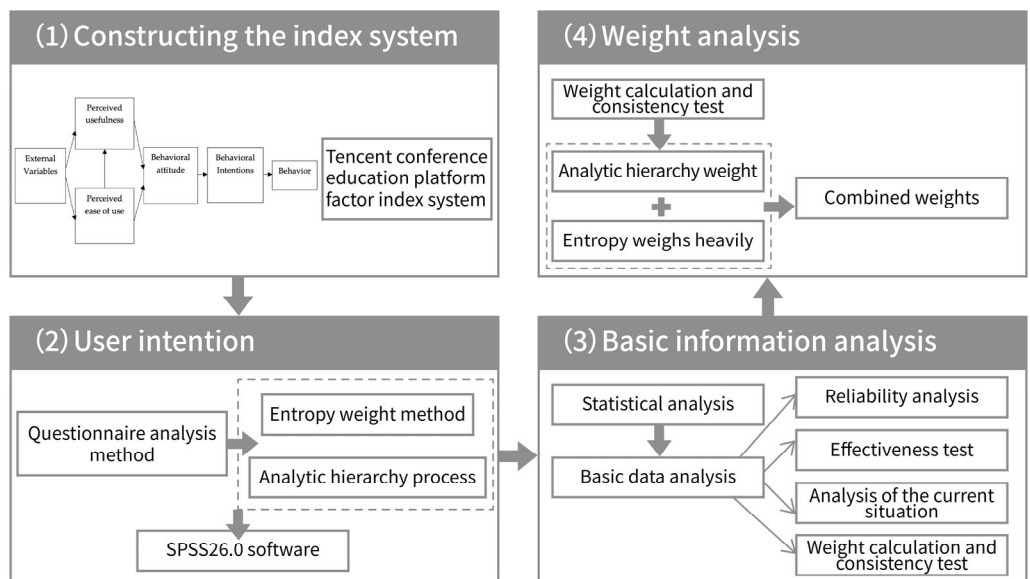

**Figure 1.** Research and analysis framework.

### 3.1. Technology Acceptance Model

Combined with international research progress, the technology acceptance model proposed by DAVIS et al. (Figure 2) was chosen as the basic framework of the study [34]. The model has received extensive attention and application from various scholars since it was proposed. Scholars fully acknowledge it as one of the most influential models for online delivery [20]. Currently, TAM is widely used and practiced in academia.

The original TAM consisted of five parts [35]. In the process of TAM being applied in various fields, researchers updated the TAM. The TAM gradually developed into TAM2 and TAM3 [36], among others. In the field of online delivery, TAM has been used by scholars in research cases to predict learning behaviors and attitudes in online environments. Tao, D et al. proposed an innovative model combining TAM and TTF to allow college students to gain online mastery in a precise environment within a particular time frame. Almulla, M.A et al. reviewed and extended TAM to assess the capacity and role of online education. They studied some stages of COVID-19 popularity and argued that digital applied sciences should be used in trainers. Saleh, S.S et al. used TAM as a model to study quantitative analysis to explore the elements of users' stance on using online lectures. TAM has been validated as a generic framework by researchers. Currently, frameworks include original models [37] or updated models [38]. Models are used for basic frameworks in different disciplines to demonstrate the accuracy and validity of the technique [39,40]. The

usage attitudes of the Chinese participants of the Tencent meeting platform are studied. Also, factors such as the external environment are studied in TAM. Based on previous theories, the participants' satisfaction was enhanced. The research team used an open-ended questionnaire [41] and mathematical statistics to conduct a practical study. We analyzed the factors influencing participants' satisfaction.

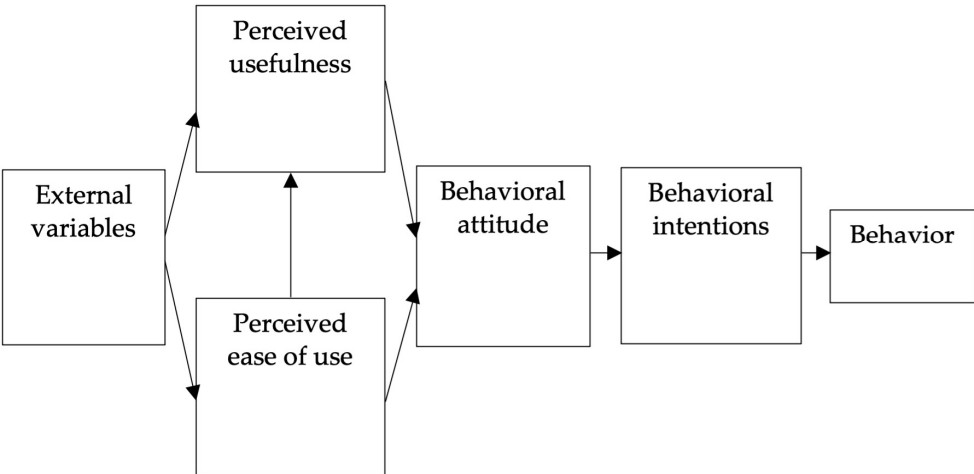

**Figure 2.** Technology acceptance model.

### 3.2. Analytical Hierarchy Process

In the early 1970s, Toma at the University of Pittsburgh proposed a multi-criteria analysis method, the analytic hierarchy process (AHP) [42]. AHP is a decision analysis tool designed to deal with complex problems through case-by-case analysis. It combines quantitative and qualitative analysis to determine the weight or priority of individual factors [43]. The basic idea is to divide the complex decision problem into a hierarchical structure to solve it, and qualitative and quantitative criteria are used in the decision-making process. In the range of 1 to 9, the indicators are compared and evaluated two by two, and the indicators form the comparative components and matrix of weights [44].

### 3.3. Entropy Method

In an actual evaluation process, the personal preferences and subjective judgments of platform participants influence the indicator weights and lead to the final evaluation results that are not reliable [45]. The entropy weighting method was used in the study of the influencing factors of online teaching platforms. It can not only calculate the indicator weights efficiently and improve the accuracy of data [46] but also can provide a favorable basis for the influence factors of multiple indicators and avoid the influence of the subjective judgment errors of the participants on the weights [47].

We use entropy weighting techniques to determine the combined weights of primary and tertiary indicators for the Tencent meeting platform. A standardized weight matrix was constructed to calculate the approximation and desirability of the target objects. The comprehensive weight of each indicator is calculated by entropy value and weight, and its importance is judged by this criterion. The top-ranked influencing factors are taken as the best optimization direction.

### 3.4. Questionnaire Design

Based on TAM (technology acceptance model) as the basic framework, based on the preliminary analysis of the questionnaire survey results and the in-depth understanding of the Tencent conference user experience, the paper constructs a model of the influencing factors of Tencent meeting usage behavior. The model covers key factors such as the perceived usability, perceived ease of use, external environmental impact, platform satisfaction, and willingness to continue using the platform.

The sources of each indicator include three main aspects. First, to revise the test indicators that have been widely used in the literature review. Second, the index system was revised with reference to international authorities but for lesser use by Chinese participants. Third, the index scale was constructed based on the specific contexts and relevant research theories that fit the Tencent conference online delivery platform. The measure of B1 draws on the measures proposed by Davis [48] and Bhattacherjee [49]. B2 was drawn from the form of measurement proposed by Davis (1989) [34] and others, while B3 draws on the measures proposed by Davis (1989) [34] and Venkatesh and Davis et al. (2000) [50]. The measure for B4 draws on the measures proposed by Lee [51]. And B5 draws on the measures proposed by Yin et al. [52]. This led to the establishment of the Tencent meeting online delivery platform factor indicator system (Table 1).

**Table 1.** Tencent conference web delivery platform factor index system.

| Primary Indicators | Secondary Indicators | Tertiary Indicators | Literature Sources |
|---|---|---|---|
| B1 Perceived usefulness | C1 Learning resources | D1 Effective sharing of learning materials<br>D2 Instant cloud recording of courses<br>D3 Applicable to different disciplines | Hu, Y.; et al. (2021) [47] |
| | C2 Course study | D4 Enriching learning content<br>D5 Improve learning efficiency<br>D6 Management Learning Program | |
| | C3 Feedback | D7 Classroom response feedback on learning behaviors | |
| B2 Perceived ease of use | C4 Convenience<br>C5 Degree of stability<br>C6 Learning timeframe | D8 Convenience of operation<br>D9 Proficiency in use<br>D10 No geographical time difference | Mayer, R. E. (2019) [10]<br>Davis, F. D. (1989) [34] |
| B3 External environment | C7 Software services | D11 System completeness<br>D12 Interactive latency | Davis, F. D. (1989) [34]<br>Bhattacherjee, A. (2001) [49]<br>Venkatesh, V.; et al. (2000) [50] |
| | C8 Platform support | D13 Good learning atmosphere support<br>D14 User online communication support | |
| B4 Platform satisfaction | C9 Use effect | D15 Test classroom effectiveness with online tests<br>D16 As opposed to offline classes<br>D17 Platform use satisfaction | Yang, H. H.; et al. (1997) [19]<br>Davis, F. D. (1989) [34] |
| | C10 Learning experience | D18 Cloud recording effect<br>D19 Shared screen effect<br>D20 Learning Interaction Effect | |
| B5 Willingness to continue using | C11 Degree of use | D21 Degree of frequency of use | Joshi, O.; et al. (2020) [32]<br>Lee, M.-C. (2010) [51] |
| | C12 Recommendation level | D22 Willingness to recommend others to use | |

The questionnaire in general consists of two sections. In the first part, the demographic characteristics of the main survey sample, such as gender and education level, are investigated. The second part, which is the core part of the questionnaire, mainly investigates the participants' acceptance of the Tencent meeting platform. There are 22 measurement items in this part (Table 2), B1, B2, B3, B4, and B5 have 7, 3, 4, 6, and 2 measurement items, respectively. The questionnaire was in the form of a scale [53]. We used a nine-level Likert scale to measure respondents' attitudes, with increasing importance from 1 to 9, with 9 representing very important, and all measurement items were single-choice questions.

**Table 2.** Description of the indicator conversion questionnaire.

| Tertiary Indicators | Indicator Description |
|---|---|
| D1 Effective sharing of learning materials | Effective sharing of learning materials for you? |
| D2 Instant cloud recording of courses | Instant cloud recording sessions for you? |
| D3 Applicable to different disciplines | Applicable to different disciplines of study for you? |
| D4 Enriching learning content | Can enrich the learning content for you? |
| D5 Improve learning efficiency | Can improve the efficiency of learning for you? |
| D6 Management Learning Program | Able to manage a study program for you? |
| D7 Classroom response feedback on learning behaviors | Classroom responses to feedback on learning behaviors for you? |
| D8 Convenience of operation | How easy is it for you to operate? |
| D9 Proficiency in use | How proficient you think you are in using the Tencent meeting platform. |
| D10 No geographical time difference | You think the Tencent meeting platform is not limited by geographical time difference. |
| D11 System completeness | System perfection for you? |
| D12 Interactive latency | Interactive latency for you? |
| D13 Good learning atmosphere support | Good learning atmosphere provided for you? |
| D14 User online communication support | What does the provision of online user communication mean to you? |
| D15 Test classroom effectiveness with online tests | Learning through online test detection for you? |
| D16 As opposed to offline classes | Satisfaction compared to offline classes? |
| D17 Platform use satisfaction | How satisfied are you with using Tencent's meeting platform? |
| D18 Cloud recording effect | Satisfaction with cloud recording results? |
| D19 Shared screen effect | Satisfaction with shared screen effects? |
| D20 Learning Interaction Effect | Satisfaction with the effectiveness of learning interactions? |
| D21 Degree of frequency of use | How often will you use Tencent meetings in the future? |
| D22 Willingness to recommend others to use | Would you recommend Tencent conferences to others? |

Questionnaire Survey

The questionnaire was distributed online using the online questionnaire tool "Questionnaire.com (http://www.sojump.com/, accessed on 1 July 2022)". The research team forwarded the link to Chinese online social media platforms such as WeChat, Weibo, and QQ. The questionnaires were distributed from 5 July to 15 July 2022. A total of 320 questionnaires

were distributed and 320 questionnaires were collected. The questionnaires that selected the same options all over or those that had no experience in using Tencent meeting platform were excluded. The remaining number of valid questionnaires was 297. The questionnaire recovery rate and effective rate were 100% and 93.65%, respectively.

The data results of the questionnaire survey were analyzed by SPSS 26.0 software, and the main process was as follows. First, the user characteristics and related statistics were analyzed. Secondly, the credibility and reliability of the online lecture platform were studied through credibility analysis and status quo analysis. The influence effects of the use of online lecture platforms were analyzed by combining hierarchical analysis and entropy power method. Finally, the influence of Chinese participants on the online lecture platform was analyzed.

Data were downloaded from the questionnaire website, screened, and entered into SPSS 26.0 for sample distribution analysis. The characteristics of the participants in terms of gender, age, and education were mainly investigated (Table 3). The data showed that 31.3% of the participants were male and 68.7% were female. In terms of age, 3.0% were under 18 years old and 81.8.% were 18–30 years old, 6.7% of the survey respondents were 31–40 years old while 7.4% of them were 41–50 years old. Only one percent of the participants were beyond 51 years old. It is worth mentioning that the age group is mainly concentrated in 18–30 years old. When it comes to the education background, the percentages of the respondents who were undergraduate and graduate students were 69.7% and 19.9%, respectively, while the percentages of respondents who had received junior high school and below, high school, and specialist education were 1.0%, 3.0%, and 6.4%, respectively. The surveyed sample profile is generally consistent with the group of students who use online platforms to participate in learning.

**Table 3.** Statistical analysis of questionnaire description.

|  |  | Frequency | Percentage/% |
|---|---|---|---|
| Gender | Men | 93 | 31.3 |
|  | Women | 204 | 68.7 |
| Age | Less than 18 | 9 | 3.0 |
|  | 18~30 | 243 | 81.8 |
|  | 31~40 | 20 | 6.7 |
|  | 41~50 | 22 | 7.4 |
|  | 51 or more | 3 | 1.0 |
| Educational Background | Junior high school and below | 3 | 1.0 |
|  | High school | 9 | 3.0 |
|  | Specialty | 19 | 6.4 |
|  | Undergraduate | 207 | 69.7 |
|  | Graduate students | 59 | 19.9 |

## 4. Data Analysis

### 4.1. Reliability Analysis

We used Cronbach's $\alpha$ coefficient to check the reliability of each stratum. Reliability detects the internal consistency of the scales and determines the reliability of the questionnaire. We mainly test the stability and reliability of the questionnaire results. It is generally believed that Cronbach's $\alpha$ coefficient above 0.8 represents a high degree of confidence. The questionnaire data were imported into SPSS 26.0 for reliability analysis. The Cronbach $\alpha$ coefficient was 0.939, which is higher than 0.8, indicating that the reliability quality of the probe data is high (Table 4). The reliability of the selected scales was relatively satisfactory, and indicators had good internal consistency.

**Table 4.** Analysis results.

| Sample | Number of Samples | Cronbach $\alpha$ Numerical Value |
|---|---|---|
| 22 | 297 | 0.939 |

### 4.2. Effectiveness Test

The accuracy and reasonableness of the factors were determined by validity testing. Factor analysis is the main analysis method of the validity test. The value of KMO can be used to judge the level of fitness of the data extracted. The level of information extraction is stated according to the variance interpretation rate. The research team demonstrates the degree of relationship between the question under test and the indicator by the factor loading coefficient. The validity test calculated the KMO value of 0.941, which is higher than 0.6 (Table 5). The explained values of the variance of the factors were 20.869%, 19.068%, and 17.203% in that order. The approximate chi-square was 3361.916. This means that information regarding the research items could be extracted effectively.

**Table 5.** Effectiveness analysis.

| KMO Value | | 0.941 |
|---|---|---|
| Bartlett sphericity test | Approximate cardinality | 3361.916 |
| | df | 231 |
| | $p$ value | 0.000 |

### 4.3. Analysis of the Current Situation

The data were analyzed for status quo using SPSS 26.0 (Table 6). The mean values of the indicators from B1 to B5 were 6.400, 6.939, 6.772, 6.565, and 6.663, respectively, and the mean value was close to seven. A score of nine on the original scale indicates extremely important, very satisfactory, frequently used, and highly recommended. Through this research, we conclude that the influencing factors on using the Tencent meeting platform in the context of COVID-19 tend to be positive.

**Table 6.** Analysis of the current situation.

| | Number of Cases | Minimum Value | Maximum Value | Average Value | Standard Deviation |
|---|---|---|---|---|---|
| B1 | 297 | 3.000 | 9.000 | 6.400 | 1.218 |
| B2 | 297 | 3.670 | 9.000 | 6.939 | 1.241 |
| B3 | 297 | 3.750 | 9.000 | 6.772 | 1.117 |
| B4 | 297 | 3.330 | 9.000 | 6.565 | 1.155 |
| B5 | 297 | 2.000 | 9.000 | 6.663 | 1.331 |

### 4.4. Weight Calculation and Consistency Test

In the hierarchical analysis, the levels reflect the relationship between the elements. However, the proportion of each element in the objective scale varies in the minds of different decision-makers. Therefore, this study adopts the evaluation criteria of the analytical hierarchy process as a method to determine the relative position and distribution of project elements in the platform hierarchy model. With this approach, we were able to quantify research results and assess the importance of the design elements. The research team used mathematical tools to construct the data into matrices for processing. It is assumed that the Tencent meeting platform has n influencing factors, which are $B_1 \ldots, B_i, \ldots, B_j, \ldots$ Therefore, a two-by-two comparison is performed for each item element. The judgment matrix is as follows:

$$B = \begin{bmatrix} 1 \cdots b_{1i} \cdots b_{1j} \cdots b_{1n} \\ b_{i1} \cdots 1 \cdots b_{ij} \cdots b_{in} \\ b_{j1} \cdots b_{ji} \cdots 1 \cdots b_{jn} \\ b_{n1} \cdots b_{ni} \cdots b_{nj} \cdots 1 \end{bmatrix} = (b_{ij})_{n \times n} \tag{1}$$

By the Perron–Frobenius theorem, the matrix B has a unique non-zero eigenroot, i.e., the largest eigenroot ($\lambda_{max}$) corresponds to the eigenvector (w).

$$B_w = \lambda_{max} w \tag{2}$$

The specific steps for calculating the eigenvectors using the sum–product method are as follows:

Normalize the data in B by columns:

$$\overline{b}_{ij} = b_{ij} / \sum_{k=1}^{n} b_{kj} \tag{3}$$

Summing the above results:

$$\widetilde{w_i} = \sum_{j=1}^{n} \overline{b}_{ij} (i = 1, 2, \cdots, n) \tag{4}$$

The summed vector is divided by n to obtain the weight vector:

$$\widetilde{w_i} = \widetilde{w_i} / n \tag{5}$$

Maximum characteristic root:

$$\lambda_{max} = \frac{1}{n} \sum_{i=1}^{n} \frac{(B_w)_i}{w_i} \tag{6}$$

where $(B_w)_i$ denotes the ith component of the vector and $B_w$. denotes the first component of the vector.

Based on the above Equations (1)–(6), the weight values of the design element objectives of the primary and tertiary indicators are calculated. These results are crucial for determining the importance ranking of factors in design decisions. Inconsistencies may occur when comparing the importance of different factors. To ensure the reliability of the calculated results, we performed a consistency check on the calculated data. The consistency of the results needs to be checked, and the test procedure is as follows.

The steps of the formula for CR (consistency ratio) are as follows:

$$CR = \frac{CI}{RI} \tag{7}$$

In Equation (7), RI denotes the average consistency index given by hierarchical analysis, $\lambda_{max}$ denotes the maximum eigenvalue in the matrix, and i denotes the judgment matrix order.

CI (Consistency index) is calculated as follows:

$$CI = \frac{\lambda_{max} - 4}{4 - 1} \tag{8}$$

According to Equation (8), when the consistency ratio CR value is less than or equal to 0.1, it indicates that the data pass the consistency test, and the degree of inconsistency is within the permissible range. If the CR value exceeds 0.1, it means that the result does not pass the consistency test, and previous evaluation data should be corrected. Therefore, the research team tested the primary indicators B1, B2, B3, B4, and B5, and tertiary indicators of this study. The results show that the data all passed the consistency test. Then, the combined weights of each element were calculated (Tables 7–13).

**Table 7.** Weight values of level 1 indicators for the study of influencing factors of Tencent conference online delivery platform.

|  | B1 | B2 | B3 | B4 | B5 | $w_i$ | $\lambda_{max}$ | CI | CR |
|---|---|---|---|---|---|---|---|---|---|
| B1 | 1 | 0.922 | 0.945 | 0.975 | 0.96 | 3.081 | | | |
| B2 | 1.084 | 1 | 1.025 | 1.057 | 1.041 | 1.432 | | | |
| B3 | 1.058 | 0.976 | 1 | 1.032 | 1.016 | 1.863 | 22.000 | 0.000 | 0.000 |
| B4 | 1.026 | 0.946 | 0.969 | 1 | 0.985 | 2.709 | | | |
| B5 | 1.041 | 0.96 | 0.984 | 1.015 | 1 | 0.916 | | | |

**Table 8.** Judgment matrix and weight values for "perceived usefulness".

| B1 | D1 | D2 | D3 | D4 | D5 | D6 | D7 | $w_i$ | $\lambda_{max}$ | CI | CR |
|---|---|---|---|---|---|---|---|---|---|---|---|
| D1 | 1 | 0.929 | 0.947 | 0.96 | 0.998 | 1.041 | 1.002 | 0.140 | | | |
| D2 | 1.076 | 1 | 1.019 | 1.033 | 1.074 | 1.121 | 1.078 | 0.151 | | | |
| D3 | 1.056 | 0.982 | 1 | 1.014 | 1.055 | 1.1 | 1.058 | 0.148 | | | |
| D4 | 1.041 | 0.968 | 0.986 | 1 | 1.04 | 1.084 | 1.043 | 0.146 | 7.000 | 0.000 | 0.000 |
| D5 | 1.002 | 0.931 | 0.948 | 0.962 | 1 | 1.043 | 1.003 | 0.140 | | | |
| D6 | 0.96 | 0.892 | 0.909 | 0.922 | 0.959 | 1 | 0.962 | 0.135 | | | |
| D7 | 0.998 | 0.928 | 0.945 | 0.959 | 0.997 | 1.04 | 1 | 0.140 | | | |

**Table 9.** Judgment matrix and weighting values for "perceived ease of use".

| B2 | D8 | D9 | D10 | $w_i$ | $\lambda_{max}$ | CI | CR |
|---|---|---|---|---|---|---|---|
| D8 | 1 | 1.013 | 0.973 | 0.332 | | | |
| D9 | 0.987 | 1 | 0.96 | 0.327 | 3.000 | 0.000 | 0.000 |
| D10 | 1.028 | 1.042 | 1 | 0.341 | | | |

**Table 10.** Judgment matrix and weighting values for "external environment".

| B3 | D11 | D12 | D13 | D14 | $w_i$ | $\lambda_{max}$ | CI | CR |
|---|---|---|---|---|---|---|---|---|
| D11 | 1 | 1.031 | 1.033 | 1.02 | 0.255 | | | |
| D12 | 0.97 | 1 | 1.003 | 0.99 | 0.248 | | | |
| D13 | 0.968 | 0.997 | 1 | 0.987 | 0.247 | 4.000 | 0.000 | 0.000 |
| D14 | 0.981 | 1.011 | 1.013 | 1 | 0.250 | | | |

**Table 11.** Judgment matrix and weighting values for "platform satisfaction".

| B4 | D15 | D16 | D17 | D18 | D19 | D20 | $w_i$ | $\lambda_{max}$ | CI | CR |
|---|---|---|---|---|---|---|---|---|---|---|
| D15 | 1 | 1.048 | 0.965 | 0.968 | 0.944 | 0.956 | 0.163 | | | |
| D16 | 0.954 | 1 | 0.921 | 0.923 | 0.901 | 0.912 | 0.156 | | | |
| D17 | 1.036 | 1.086 | 1 | 1.003 | 0.978 | 0.991 | 0.169 | | | |
| D18 | 1.033 | 1.083 | 0.997 | 1 | 0.975 | 0.988 | 0.169 | 6.000 | 0.000 | 0.000 |
| D19 | 1.059 | 1.11 | 1.022 | 1.025 | 1 | 1.013 | 0.173 | | | |
| D20 | 1.046 | 1.096 | 1.009 | 1.012 | 0.987 | 1 | 0.171 | | | |

**Table 12.** Judgment matrix and weighting values for "intention to continue using".

| B5 | D21 | D22 | $w_i$ | $\lambda_{max}$ | CI | CR |
|---|---|---|---|---|---|---|
| D21 | 1 | 1.009 | 0.502 | | | |
| D22 | 0.991 | 1 | 0.498 | 2.000 | 0.000 | 0.000 |

**Table 13.** Combined weight value of the elements of the three-level indicator target.

| Primary Indicators | Weight Values of Primary Indicators | Tertiary Indicators | Combined Weight Value of the Three Levels of Indicators $w_i$ |
|---|---|---|---|
| B1 | 0.3081 | D1 | 0.0432 |
| | | D2 | 0.0465 |
| | | D3 | 0.0456 |
| | | D4 | 0.0450 |
| | | D5 | 0.0432 |
| | | D6 | 0.0415 |
| | | D7 | 0.0431 |
| B2 | 0.1432 | D8 | 0.0475 |
| | | D9 | 0.0469 |
| | | D10 | 0.0488 |
| B3 | 0.1863 | D11 | 0.0475 |
| | | D12 | 0.0461 |
| | | D13 | 0.0460 |
| | | D14 | 0.0466 |
| B4 | 0.2709 | D15 | 0.0442 |
| | | D16 | 0.0422 |
| | | D17 | 0.0458 |
| | | D18 | 0.0457 |
| | | D19 | 0.0468 |
| | | D20 | 0.0462 |
| B5 | 0.0916 | D21 | 0.0460 |
| | | D22 | 0.0456 |

First, the combined weight values of the elemental targets of the three-level indicators were tested for consistency (Table 13). The operation procedures and results are as follows:

$$CI = \sum_{j=1}^{m} b_j CI_j = 0.000 \tag{9}$$

$$CR = \frac{CI}{RI} = 0.000 < 0.1 \tag{10}$$

Second, the conclusion is drawn from Equations (9) and (10) $CR = \frac{CI}{RI} = 0.000 < 0.1$. The hierarchical total ranking of matrix B is consistent with the principle of consistency test. The calculation results of the integrated weight values of the three levels of index elements are scientific and reasonable. The calculation results can effectively guide the research practice of impact factors. After data calculation, it is concluded that (1) the weight percentages of the primary indicators are 0.3081, 0.1432, 0.1863, 0.2709, and 0.0916, respectively. (2) The weight shares of the tertiary indicators D1–D22 in the primary indicator B ranged from 0.135 to 0.332, and the results are shown in Tables 8–12. (3) The weight values of B1 and D1–D7, B2 and D8–D10, B3 and D11–D14, B4 and D15–20, and B5 and D21–D22 were multiplied correspondingly to obtain the combined weight values as shown in Table 13. By comparing the combined weight values $w_i$ of the three levels of indicators in Table 13, we ranked the combined demand importance of the factors affecting the Tencent meeting users (Figure 3).

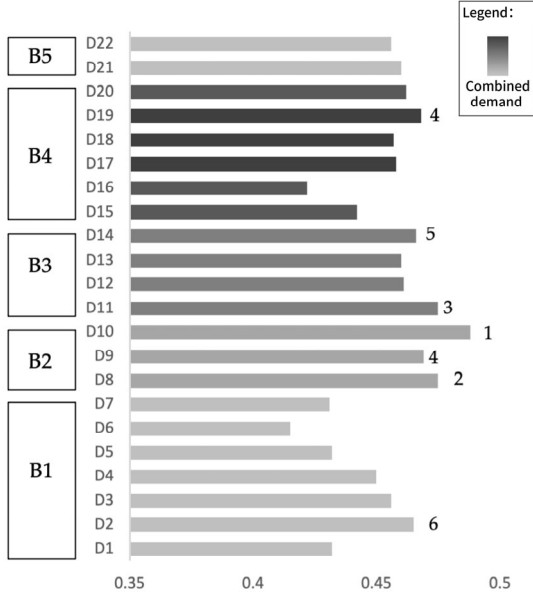

**Figure 3.** Tencent conference influence factor comprehensive weighting chart.

*4.5. Determination of Index System Weights Based on Entropy Weighting Factor*

The entropy weighting method is a target task method. It yields more accurate weights compared to the subjective challenge method. Entropy is a measure of the degree of disorder in a system. By measuring the degree of disorder of a variable, the amount of information possessed by the variable can be compared to obtain the weight of the indicator variable. The entropy weighting method first calculates the entropy weight of an indicator by applying information entropy to standardize the raw data. In this case, if the variable is a high-quality indicator, the standardization system for its value $Y_{ij}$ is as follows:

$$Y_{ij} = \frac{X_{ij} - X_{i_{min}}}{X_{i_{max}} - X_{i_{min}}} \tag{11}$$

If the stratum represents a negative indicator, the normalization formula is as follows:

$$Y_{ij} = \frac{X_{i_{min}} - X_{ij}}{X_{i_{max}} - X_{i_{min}}} \tag{12}$$

In the above Equations (11) and (12), $X_{i_{max}}$ and $X_{i_{min}}$ correspond to the maximum and minimum values in the hierarchy, respectively. In turn, $Y_{ij}$ is the normalized result of setting the jth prevention and control impact factor, where $E_i$ is the entropy price of the jth indicator, n is the number of comparison indicators, and ln is the natural logarithm function. The entropy value $e_j$ of the jth stratum is determined.

$$e_j = -\frac{1}{\ln m} \sum_{i=1}^{m} P_{ij} \ln P_{ij} \tag{13}$$

$$P_{ij} = \frac{Y_{ij}}{\sum_{i=1}^{m} Y_{ij}} \tag{14}$$

Based on Equations (13) and (14), the following conclusions were calculated. The information entropy values of the first-level indicators were 0.9968, 0.9971, 0.9976, 0.9973, and 0.9964, respectively (Table 14). The information entropy values of the tertiary indicators are listed in Table 15.

**Table 14.** Results of the weighting of the first-level indicators based on the entropy weighting method.

| Primary Indicators | Information Entropy Value $e_j$ | Information Utility Value | Weighting Factor $w_j$ |
|---|---|---|---|
| B1 | 0.9968 | 0.0032 | 0.2163 |
| B2 | 0.9971 | 0.0029 | 0.1944 |
| B3 | 0.9976 | 0.0024 | 0.1624 |
| B4 | 0.9973 | 0.0027 | 0.1842 |
| B5 | 0.9964 | 0.0036 | 0.2426 |

**Table 15.** Results of the weighting of the three-level indicators based on the entropy weighting method.

| Tertiary Indicators | Information Entropy Value $e_j$ | Information Utility Value | Weighting Factor $w_j$ |
|---|---|---|---|
| D1 | 0.9937 | 0.0063 | 0.0554 |
| D2 | 0.9941 | 0.0059 | 0.0522 |
| D3 | 0.9951 | 0.0049 | 0.0430 |
| D4 | 0.9945 | 0.0055 | 0.0490 |
| D5 | 0.9940 | 0.0060 | 0.0536 |
| D6 | 0.9929 | 0.0071 | 0.0626 |
| D7 | 0.9936 | 0.0064 | 0.0565 |
| D8 | 0.9956 | 0.0044 | 0.0388 |
| D9 | 0.9956 | 0.0044 | 0.0386 |
| D10 | 0.9957 | 0.0043 | 0.0385 |
| D11 | 0.9960 | 0.0040 | 0.0354 |
| D12 | 0.9945 | 0.0055 | 0.0488 |
| D13 | 0.9953 | 0.0047 | 0.0415 |
| D14 | 0.9956 | 0.0044 | 0.0390 |
| D15 | 0.9944 | 0.0056 | 0.0494 |
| D16 | 0.9924 | 0.0076 | 0.0676 |
| D17 | 0.9959 | 0.0041 | 0.0364 |
| D18 | 0.9950 | 0.0050 | 0.0447 |
| D19 | 0.9959 | 0.0041 | 0.0367 |
| D20 | 0.9965 | 0.0035 | 0.0314 |
| D21 | 0.9955 | 0.0045 | 0.0402 |
| D22 | 0.9954 | 0.0046 | 0.0405 |

Based on the calculated information entropy of each factor $e_1$, $e_2$, ..., $e_k$, the weights $w_j$ of each factor are calculated, and the formula is as follows:

$$w_j = \frac{1 - e_j}{k - \sum_{j=1}^{k} e_j} \qquad (15)$$

The results of the specific values of each level were calculated with the Formula (15) according to the entropy weighting method. The weight coefficients $w_j$ of the first-level indicators were calculated as 0.2163, 0.1944, 0.1624, 0.1842, and 0.2426, respectively (Table 14). The weight coefficients $w_j$ of the tertiary indicators are shown in Table 15.

*4.6. Combined Weights*

Based on the results of the data from the above two methods of AHP and entropy weighting for each level, the combined weights $C_j$ are calculated.

$$C_j = \frac{w_i w_j}{\sum_{j=1}^{n} w_i w_j} \qquad (16)$$

Equation (16) of $w_i$ and $w_j$ represents the evaluation index weights calculated by the AHP and entropy weighting method. The subjective and objective assignments of each index are calculated comprehensively. After calculation, the combined weights $C_j$ of the

primary indicators and the weights $w_j$ are listed in Table 16. The combined weights $C_j$ of the tertiary indicators and entropy weighting method weights $w_j$ are shown in Table 17.

**Table 16.** Two weighting methods and comprehensive weighting results of first-level indicators.

| Primary Indicators | AHP Weights $w_i$ | Entropy Method Weights $w_j$ | Combined Weights $C_j$ |
|---|---|---|---|
| B1 | 0.3081 | 0.2163 | 0.3385 |
| B2 | 0.1432 | 0.1944 | 0.1414 |
| B3 | 0.1863 | 0.1624 | 0.1537 |
| B4 | 0.2709 | 0.1842 | 0.2535 |
| B5 | 0.0916 | 0.2426 | 0.1129 |

**Table 17.** Two weighting methods and comprehensive weighting results of the three-level indicators.

| Tertiary Indicators | AHP Weights $w_i$ | Entropy Method Weights $w_j$ | Combined Weights $C_j$ |
|---|---|---|---|
| D1 | 0.0432 | 0.0554 | 0.0530 |
| D2 | 0.0465 | 0.0522 | 0.0538 |
| D3 | 0.0456 | 0.0430 | 0.0434 |
| D4 | 0.0450 | 0.0490 | 0.0489 |
| D5 | 0.0432 | 0.0536 | 0.0513 |
| D6 | 0.0415 | 0.0626 | 0.0576 |
| D7 | 0.0431 | 0.0565 | 0.0540 |
| D8 | 0.0475 | 0.0388 | 0.0408 |
| D9 | 0.0469 | 0.0386 | 0.0401 |
| D10 | 0.0488 | 0.0385 | 0.0416 |
| D11 | 0.0475 | 0.0354 | 0.0373 |
| D12 | 0.0461 | 0.0488 | 0.0498 |
| D13 | 0.0460 | 0.0415 | 0.0423 |
| D14 | 0.0466 | 0.0390 | 0.0403 |
| D15 | 0.0442 | 0.0494 | 0.0484 |
| D16 | 0.0422 | 0.0676 | 0.0632 |
| D17 | 0.0458 | 0.0364 | 0.0369 |
| D18 | 0.0457 | 0.0447 | 0.0453 |
| D19 | 0.0468 | 0.0367 | 0.0381 |
| D20 | 0.0462 | 0.0314 | 0.0321 |
| D21 | 0.0460 | 0.0402 | 0.0410 |
| D22 | 0.0456 | 0.0405 | 0.0410 |

The results of the combined weight $C_j$ of the first-level indicators are shown in Table 16 and Figure 4. The hierarchical analysis weight $w_i$ of B5 and the combined weight $C_j$ of the first-level indicators are the smallest, 0.0916 and 0.1129, respectively. The entropy weight $w_i$ is 0.2426. The ranking of the importance of the user-influencing factors from lowest to highest is B5 (continued willingness to use), B2 (perceived ease of use), B3 (external environment), B4 (platform satisfaction), and B1 (perceived usefulness).

The results of the combined weights of the three levels of indicators $C_j$ are shown in Table 17 and Figure 5. According to the calculation results, the influence of the combined weights of the three levels of indicators is ranked from highest to lowest as D6, D7, D2, D1, D5, D12, D4, D15, D18, D3, D13, D10, D21, D22, D8, D14, D9, D19, D11, D17, and D20. Thus, the factors that have a relatively large impact on willingness to use consistently (B5) are the degree of frequency of use (D21) and willingness to recommend others to use (D22), and the combined weight $C_j$ of the two accounts for 0.082.

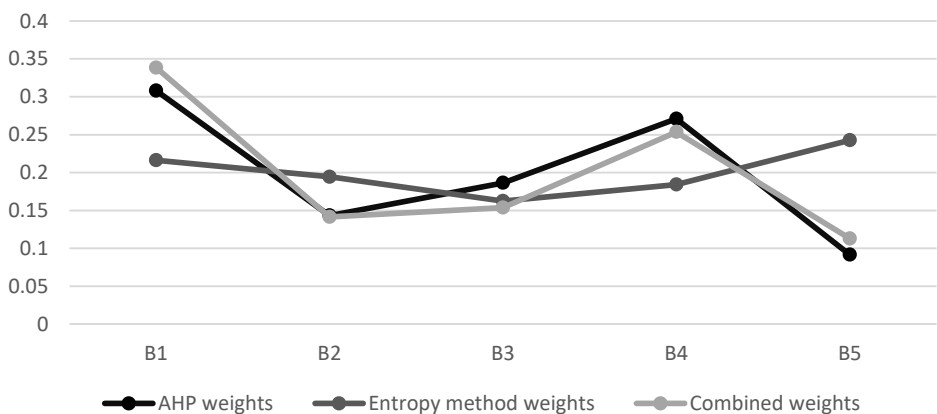

**Figure 4.** Two weighting methods and comprehensive weight ranking of first-level indicators.

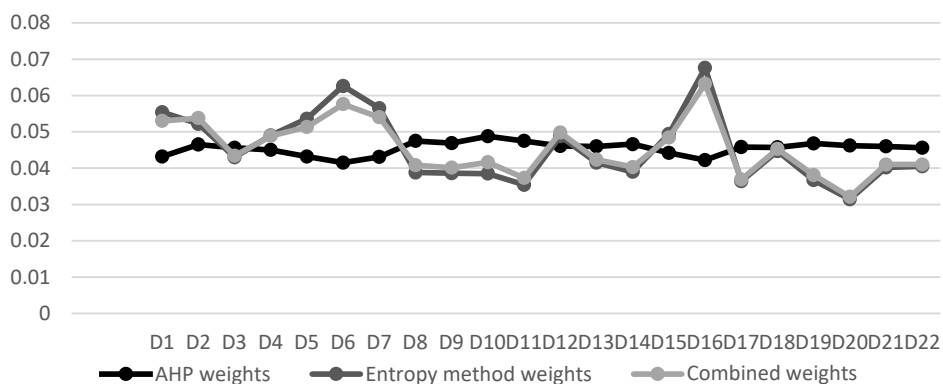

**Figure 5.** Two weighting methods and comprehensive weight ranking of the three-level indicators.

### 5. Discussion

Focusing on the Tencent conference software, this paper investigates and analyzes the user behavior and preferences of online lecture platforms in the context of sustainable education. Based on this research background, the following conclusions are drawn.

First, the perceived ease of use (B2) is the most important factor influencing the users to use Tencent's meeting delivery platform. Among the various factors that make up the perceived ease of use (B2), no time and space constraints (D10) is the key factor that influences people's judgment of the perceived ease of use. The weight value of no time and space constraints (D10) is 0.0488, ranking first among the weight values of the three-level indicators. Tencent conference breaks through the time and space constraints of traditional offline education methods and provides the convenience of instant participation. Mäkelä et al. similarly concluded that the key factors affecting the perceived ease of the use of online teaching are factors such as breaking the constraints, resource sharing, and timely feedback. Sustainable development in education means consolidating and developing a good situation in education, and being realistic and innovative.

Second, perceived usefulness (B1) can have a significant impact on the participants' intention to use the online delivery platform. The composite weight value of perceived usefulness (B1) was 0.3385. Among the multiple factors constituting perceived usefulness, the relatively high composite weights of study schedule management (D6) and classroom response feedback on learning behaviors (D7) indicated that the schedule and planning setup prior to using Tencent sessions largely influenced the participants' evaluation of the usefulness of online education. The composite weight values of D6 (managing the study schedule) and D7 (classroom response feedback on learning behaviors) had the combined weight values of 0.0576 and 0.0540, respectively. Teachers should actively adopt the concept of sustainable development and become the key force in promoting sustainable education. At the same time, as the successors of the future, young students shoulder the primary responsibility for promoting sustainable development.

Third, platform satisfaction (B4) had a greater impact on the participants' behavior in using online teaching platforms, with the combined weight value of "relative to offline teaching" (D16) having the greatest impact on platform satisfaction (B4). The composite weight value of platform satisfaction (B4) was 0.2535. in this case, the composite weight value of "relative to offline teaching" (D16) was 0.0632. Many participants lack experience in online teaching, which presents significant challenges to this emerging teaching model. To realize the sustainable development of vocational education, we must design both practical and efficient talent training programs. We must improve online teaching platforms because of their ability to provide instant, rapid feedback, and other advantages, so that users tend to give a more positive evaluation.

Fourth, the effect of willingness to use consistently (B5) on the participants' use of the online lecture platform was not significant, and the difference in the combined weights of the frequency of use (D21) and willingness to recommend others to use (D22) was not significant. Sustainable development is a comprehensive concept involving all aspects of society, and education is also a key factor in sustainable development. To implement the strategy of the sustainable development of Chinese education, it is necessary to establish a comprehensive and correct view of educational development.

Fifth, the interviewees' personal evaluation involves the rapid growth of educational online platforms and the key role they play in respondents' lives. As an educator, I was initially apprehensive about the sudden change, but soon became attracted to the potential and flexibility that online education offered. Virtual classrooms can transcend geographical limitations and provide students with resources and lecturers from around the world. However, technical issues, student engagement, and differences in home environments also pose challenges. The respondents' overall assessments of the experience were mixed, ranging from frustration with current limitations to optimism about future possibilities.

## 6. Conclusions

The overall study in this paper reveals the influences and the interrelationships of multiple factors that affect people's use of online education platforms. The perceived ease of use, perceived usefulness, external environment, platform satisfaction, and willingness to continue to use all influence people's behavior in using online education tools. The combined weight value of platform satisfaction (B4) was calculated to be 0.2535 (Table 16). The combined weight value relative to offline lectures (D16) was 0.0632 (Table 17). The results show that the combined weight value relative to offline lectures (D16) has the greatest impact on platform satisfaction (B4). Educational resources are the foundation of the sustainable development of education, and human resources are its core. The whole process of education should always take humanism as the core, and its goal should be to cultivate capable talents.

Network teaching not only provides people with a convenient way of communication but also releases a large portion of its development potential. Considering the sustainability of education, the world today has regarded sustainable development as an international consensus, and the sustainability of education has become a common pursuit in the global education field. The process of learning and being educated runs through one's entire life journey. Without educational resources, human survival and reproduction will not be guaranteed.

Through the empirical analysis of the Tencent conference online education platform, this study directly responds to the concern of this Special Issue on the sustainability of higher education. In the context of the disruption of the traditional education models caused by COVID-19, this study reveals how online education platforms can become a key component of education for sustainable development. Through the technology acceptance model (TAM), the research team not only identified the key factors that influence the willingness to continue using online education platforms but also provided practical strategies for higher education institutions to promote the effectiveness and sustainability of distance learning.

**Author Contributions:** Conceptualization, X.W. and Y.G.; methodology, Y.G.; software, X.M.; validation, X.W., X.M. and Z.W.; formal analysis, X.M.; investigation, Z.W.; resources, X.M.; data curation, X.M.; writing—original draft preparation, X.M. and X.W.; writing—review and editing, X.W. and Y.G.; visualization, X.W.; supervision, X.W.; project administration, X.W.; funding acquisition, X.W. All authors have read and agreed to the published version of the manuscript.

**Funding:** This research was supported by the 2022 Director's Fund Project of Anhui Institute of Culture and Tourism Innovation and Development (Project No. ACTZ2022ZD01), the "Digital Intelligence Rural Culture and Tourism Research and Innovation Team" (Project No. 2022AH010022), the China Postdoctoral Science Fund Project (Project No. 2023M730017), and the Youth Project of Anhui Academy of Social Sciences (Project No. QK202301).

**Institutional Review Board Statement:** The study was conducted according to the guidelines of the Declaration of Helsinki and approved by the Institutional Review Board (or Ethics Committee) of Anhui Cultural Tourism Innovation Development Research Institute. (protocol code No.: ECACTIDRI-2023-019 and 25 March 2024).

**Informed Consent Statement:** Informed consent was obtained from all subjects involved in the study.

**Data Availability Statement:** Data are contained within the article.

**Conflicts of Interest:** The authors declare no conflict of interest.

## Abbreviations

| | |
|---|---|
| COVID-19 | Corona Virus Disease 2019 |
| QQ | Tencent qq |
| KMO | Kaiser–Meyer–Olkin |
| MOOC | Massive Open Online Course |
| TAM | Technology acceptance model |
| TTF | Task–Technology Fit |
| AHP | Analytic hierarchy process |
| SPSS | Statistical Product and Service Solutions |

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
