# Peer review of "A Study on the Factors Influencing the Sustainable Development of Education in the Context of COVID-19: Tencent Conference Online Platform"

_sustainability, doi:10.3390/su16104240_

Round 1
Reviewer 1 Report
Comments and Suggestions for Authors
Thank for you the opportunity to review your paper. This paper explored participants’ perceptions of a certain kind of online teaching/meeting software. For most readers, especially those interested in using online meeting software, this paper is a heavy read. There were several significant issues throughout that I will summarize.
First, I did not get a clear sense of the overall purpose of the research. While the study was theoretically grounded, the authors did not make a strong case for why evaluating perceptions of online software is needed. Are university instructors more likely to use the software if it is found that perceptions of the software are positive? What about business people? If this is so, the meeting software was not well-described. The reader had almost no sense of the features of the software and what the user experience was like in terms of using it (only users’ perceptions of using it). Authors must include a clear description of the program itself so readers can understand what it actually does.
Second, why are the authors interested in this particular software compared to other software that accomplish the same thing? This study almost seems like something a company would do to gather data about user experience to improve the software and turn a profit. If the authors are interested in the effectiveness of the software in creating vibrant learning experiences, they either need to more clearly justify why user perceptions are a good measure, or include addition measures of effectiveness of the software in producing a better learning/meeting environment (without a better description of the software, as stated above, this is very difficult to visualize).
Third, their analysis was extremely math-heavy. While I appreciate their attempt to conduct rigorous analysis, they also indicate that these are “preliminary analyses.” They do not appear preliminary based on the amount of metrics, equations, and tables included, many of which do not seem necessary. The Cronbach’s alphas are good, but I cannot understand what kind of factor analysis was conducted. If they performed an EFA, I would expect to see factor loadings, Eigen values, etc. If the performed a CFA, I would expect to see traditional model fit statistics (CFI, TLI, RMSEA, chi-square). Reporting one coefficient for a supposed factor analysis is not sufficient. On the whole, these analyses are weighty, saturated with tables, and values, and do not seem to connect with most readers who would be interested in this kind of study. I suggest a simplified analysis of their participant survey, perhaps an ERA (or CFA if they justify an expected factor structure using theory). These are usually preliminary analyses as part of a first step.
In sum, I simply do not connect with the overall purpose of this study, and if the authors want to reach wide audience, I suggest they discuss implications in a straightforward, purposeful way, and conduct analyses that answer a preliminary research question or two. The quality of the writing is strong, but I cannot recommend this article for publication without significant revisions.
Reviewer 2 Report
Comments and Suggestions for Authors
The manuscript “A study on the factors influencing the sustainable development of education in the context of COVID-19: Tencent Conference Online Platform” studies the TAM model as the basis to establish the index system of influencing factors of Tencent Conference online education platform. I think the study is interesting for the educational research audience, but some aspects should be revised before acceptance.
Introduction and Literature Review. Review recent studies related to the use of online platforms in the educational area at different levels. Also, establish comparisons between the educational platforms used in China and their equivalents in other countries. What can be said about YouTube? It can be considered an educational platform. It is known that today YouTube is the main reference source in the academic area. Some examples: doi: 10.1021/acs.jchemed.3c00284.
Research Methodology: It is recommended to place a diagram that summarizes the methodology developed in the study, which allows this experience to be replicated in the future with other studies.
Data Analysis: This section is very well-detailed.
Discussion: The discussion is only limited to the statistical analysis of the data. It would be interesting to include the personal appreciation of some of the respondents. To learn a little about its social and cultural context during the COVID-19 pandemic and its influence in the academic area. One of the most relevant aspects for students during the pandemic was: access to the internet, the use of computers shared with other family members, and the space not suitable for attending classes at home. Some references: 10.3390/su14084776
Please improve the Abstract and Conclusions considering all observations.
Minor observation: a list of abbreviations at the end of the document is suggested.
Comments on the Quality of English Languagen/a
Author Response
Dear Reviewer:
Thank you and the reviewers for your valuable and helpful suggestions concerning our manuscript entitled “A study on the factors influencing the sustainable development of education in the context of COVID-19: Tencent Conference Online Platform” (Article, Manuscript ID: 2947421). We have discussed and studied the reviewers' comments in detail and addressed all the issues raised by the reviewers. The manuscript has been revised accordingly and the changes are highlighted in red. Detailed responses to the reviewers' comments are as follows.
Reviewer comment: 1
Introduction and Literature Review. Review recent studies related to the use of online platforms in the educational area at different levels. Also, establish comparisons between the educational platforms used in China and their equivalents in other countries. What can be said about YouTube? It can be considered an educational platform. It is known that today YouTube is the main reference source in the academic area. Some examples: doi: 10.1021/acs.jchemed.3c00284.
[Reply] We would like to thank the reviewers for their suggestions on our introduction, and better revisions will help readers grasp the core of the research in this paper more quickly and accurately. After discussion, our team decided to revise the introduction to review the main logic of using online platforms in education fields at different levels and make it more concise and clear. The changes include: "Online education platforms independently developed by Chinese universities or jointly initiated by social entities mainly include China University MOOC, Xuetang Online, Good University Online, and Superior Course Alliance. The representative online education platforms developed and operated by Internet enterprises mainly in-clude: NetEase Cloud Class, Tencent Class, New Oriental Online, etc. At the same time, educational platforms in other countries have their own unique advantages. For example, platforms such as Khan Academy and Coursera in the United States are favored by users around the world for their rich course resources and high-quality teaching content. Fu-tureLearn in the UK has attracted many users with its flexible learning methods and diverse course content. In contrast, China's education platforms have significant advantages in terms of content richness, teaching methods, and user experience.”
Reviewer comment: 2
Research Methodology: It is recommended to place a diagram that summarizes the methodology developed in the study, which allows this experience to be replicated in the future with other studies.
[Reply] Indeed, to ensure transparency and reproducibility of research methods, it is very helpful to provide a detailed methodological chart. So that other researchers can access and replicate the study's methods. The changes include: "Specific methods and processes (Figure 1).
Figure 1. Research and analysis framework.”
Reviewer comment: 3
Data Analysis: This section is very well-detailed.
[Reply] Thank you very much for your careful review of our research work and your positive comments on the description of the data analysis section. We are committed to the highest standards of transparency and completeness in our methodological descriptions to ensure that other researchers can accurately replicate our experiments and analytical procedures. We believe that detailed data analysis is critical to understanding research findings and their validity, especially when it comes to complex or multi-stage data processing and statistical models.
Reviewer comment: 4
Discussion: The discussion is only limited to the statistical analysis of the data. It would be interesting to include the personal appreciation of some of the respondents. To learn a little about its social and cultural context during the COVID-19 pandemic and its influence in the academic area. One of the most relevant aspects for students during the pandemic was: access to the internet, the use of computers shared with other family members, and the space not suitable for attending classes at home. Some references: 10.3390/su14084776
[Reply] In the discussion section of the research paper, combining the results of statistical analysis with the participants' personal evaluations can provide a richer perspective on interpreting the data and may reveal the deeper motivations and feelings behind the data. This hybrid method can enhance the explanatory power and application value of the research results. The changes include: “Fifth, about the interviewees' personal evaluation. The rapid growth of ed-ucational online platforms and the key role they play in respondents' lives. As an educator, I was initially apprehensive about the sudden change, but soon became attracted to the potential and flexibility that online education offered. Virtual classrooms can transcend geographical limitations and provide students with resources and lecturers from around the world. However, technical issues, stu-dent engagement, and differences in home environments also pose challenges. Respondents' overall assessments of the experience were mixed, ranging from frustration with current limitations to optimism about future possibilities.”
Reviewer comment: 5
Please improve the Abstract and Conclusions considering all observations.
[Reply] Thank you for your careful review of our research and your valuable comments on the summary and conclusions. We fully agree that the summary and conclusion sections must accurately reflect all observations of the study and have already planned the following steps to make improvements:
First, enhance the presentation of key information. In the abstract, we will highlight our main findings more clearly and ensure that these findings are closely related to the research purpose and research question.
Second, strengthen impact and meaning. In the conclusion section, we will discuss the implications of the findings in more depth, including their implications for existing literature, practical applications, and future research.
Third, define the limitations and future direction. In the conclusion, we will clearly identify any potential research limitations and suggest future directions to address these limitations.
Reviewer comment: 6
Minor observation: a list of abbreviations at the end of the document is suggested.
[Reply] We fully agree that listing abbreviations at the end of the document will help readers better understand and follow the content. We will follow your suggestion to add an exhaustive list of abbreviations at the end of the article and make sure that all abbreviations used in the article are included. In addition, we also check the full text to ensure that all abbreviations are clearly explained when they first appear, and give full terms and explanations again in the abbreviation list for easy reference by readers in interdisciplinary fields. "Appendix A
COVID-19: Corona Virus Disease 2019
QQ: Tencent qq
KMO: Kaiser-Meyer-Olkin
MOOC: Massive Open Online Course
TAM: Technology Acceptance Model
TTF: Task-Technology Fit
AHP: Analytic Hierarchy Process
SPSS: Statistical Product and Service Solutions”
We have addressed all questions according to the Reviewers’ suggestion and revised the manuscript accordingly. We hope the revised manuscript can be accepted by the reviewers. Once again, thank you very much for your helpful comments and suggestions.
-------------
Yanlong Guo
Social Innovation Design Research Centre,
Anhui University, Hefei 230601,China
Phone: 00-86-152 5655 6306, E-mail: [email protected]
Specification of corrections:
- Page 1, line 11-17:“The difficulties of offline education caused by the COVID-19 epidemic are attracting increasing public attention. Although the international research on online education platforms is gradually deepening, the research on the influencing factors of Chinese users' participation in these platforms shows different results due to regional differences. Therefore, this study adopts the technology acceptance model (TAM) as the basis to build the index system of influencing factors for Tencent conference online education platform. The questionnaire design covered five dimensions, including external environment, platform satisfaction and continuous use intention.” Has been revised
- Page 1, line 22-23:“First, perceived ease of use is the most critical factor influencing users' use of online delivery platforms.” Has been revised
- Page 1, line 32-33:“In the winter of 2019, cases of the novel coronavirus have been reported in Wuhan, Hubei Province, China.” Has been revised
- Page 1, line 34-38:“The outbreak is now having a profound impact on the health, work, and education of people around the world. The United Nations 2030 Agenda for Sustainable Development is an international initiative to address increasingly pressing global challenges and promote sustainable development worldwide.” Has been revised
- Page 2, line 78-88:“Online education platforms independently developed by Chinese universities or jointly initiated by social entities mainly include China University MOOC, Xuetang Online, Good University Online, and Superior Course Alliance. The representative online education platforms developed and operated by Internet enterprises mainly include: NetEase Cloud Class, Tencent Class, New Oriental Online, etc. At the same time, educational platforms in other countries have their own unique advantages. For example, platforms such as Khan Academy and Coursera in the United States are favored by users around the world for their rich course resources and high-quality teaching content. Future Learn in the UK has attracted many users with its flexible learning methods and diverse course content. In contrast, China's education platforms have significant advantages in terms of content richness, teaching methods, and user experience.” Has been revised
- Page 2, line 93-97:“Tencent Conferences feature its high-definition video and audio calling capabilities, allowing users to interact almost without latency. Its flexible screen sharing feature enables teachers to present teaching content in real time. At the same time, with instant messaging and group discussion room functions, students can easily have group discussions or communicate one-on-one with teachers.” Has been revised
- Page 2, line 105-111:“The overall aim of this study was to explore the key factors affecting the sustainable development of education in the context of the COVID-19 pandemic, with a particular focus on the use of online conferencing software. Tencent Conference was chosen as the research object because it was widely adopted as a major tool for teaching and business communication during the epidemic. The purpose of the study was to assess user per-ceptions of the software, as user acceptance and satisfaction directly affect the continued application and effectiveness of online software in the field of education.” Has been revised
- Page 2, line 112-118:“Indeed, if users are found to have a positive attitude towards online conferencing software, this indicates that the software is more likely to be adopted by university faculty and business professionals. This is especially important for higher education institutions, as it involves curriculum design, innovation in teaching methods, and effective use of distance education resources. Furthermore, as educational models continue to evolve, understanding how these tools meet the needs of different user groups is critical to developing sustainable and inclusive pedagogical solutions.” Has been revised
- Page 2, line 120-121:“To date, academics across the globe have highlighted the importance of online teaching in terms of educational sustainability.” Has been revised
- Page 2, line 122-125:“Online teaching broadens the scope and methods of education and learning. [15] Based on digital technology, educators in Europe are upgrading their digital skills through online teaching models.” Has been revised
- Page 3, line 164-168:“Positive progress is being made in integrating the concept of sustainable development into high-quality education and online teaching in schools. At the same time, network lecture is also affected by teaching space, time, content and way. This is reflected in the fact that students often need to interact with multiple teachers and the diversity of online platforms that teachers choose.” Has been revised
- Page 3, line 203-205:“Influenced by the COVID-19 pandemic and the international political and economic environment, global student mobility has shown a new trend.” Has been revised
- Page 4, line 238-241:“Specific methods and processes (Figure 1). Figure 1. Research and analysis framework.” Has been
- Page 5, line 272-275:“Analytic hierarchy Process (AHP). [51] AHP is a decision analysis tool designed to deal with complex problems through case-by-case analysis. It combines quantitative and qualitative analysis to determine the weight or priority of individual factors. [52]” Has been revised
- Page 6, line 302-307:“Based on TAM (Technology Acceptance Model) as the basic framework, based on the preliminary analysis of the questionnaire survey results and the in-depth understanding of Tencent conference user experience. The paper constructs a model of influencing factors of Tencent meeting usage behavior. The model covers key factors such as perceived usability, perceived ease of use, external environmental impact, platform satisfaction, and willingness to continue using the platform.” Has been revised
- Page 10, line 405-409:“Therefore, this study adopts the evaluation criteria of analytic hierarchy process as a method to determine the relative position and distribution of project elements in the platform hierarchy model. With this approach, we were able to quantify research results and assess the importance of design elements. The research team used mathematical tools to construct the data into matrices for processing.” Has been revised
- Page 10, line 430-433:“These results are crucial for determining the importance ranking of factors in design decisions. Inconsistencies may occur when comparing the importance of different factors. In order to ensure the reliability of the calculated results, we performed a consistency check on the calculated data.” Has been revised
- Page 16, line 564-567:“Focusing on Tencent conference software, this paper investigates and analyzes the user behavior and preferences of online lecture platform in the context of sustainable education. Based on this research background, the following conclusions are drawn.” Has been revised
- Page 17, line 598-602:“Teachers should actively adopt the concept of sustainable development and become the key force in promoting sustainable education. At the same time, as the successors of the future, young students shoulder the primary responsibility for promoting sustainable development.” Has been
- Page 17, line 607-612:“Many participants lack experience in online teaching, which presents significant challenges to this emerging teaching model. To realize the sustainable development of vocational education, we must design both practical and efficient talent training programs. Online teaching platform because of its ability to provide instant, rapid feedback and other advantages, so that users tend to give a more positive evaluation.” Has been revised
- Page 17, line 620-628:“Fifth, about the interviewees' personal evaluation. The rapid growth of educational online platforms and the key role they play in respondents' lives. As an educator, I was initially apprehensive about the sudden change, but soon became attracted to the potential and flexibility that online education offered. Virtual classrooms can transcend geographical limitations and provide students with resources and lecturers from around the world. However, technical issues, student engagement, and differences in home environments also pose challenges. Respondents' overall assessments of the experience were mixed, ranging from frustration with current limitations to optimism about future possibilities.” Has been revised
- Page 17, line 637-640:“Educational resources are the foundation of sustainable development of education, and human resources are its core. The whole process of education should always take humanism as the core, and its ultimate goal is to cultivate capable talents.” Has been revised
- Page 17-18, line 641-673:“Network teaching not only provides people with a convenient way of communication, but also releases the huge space of its development potential. Considering the sustainability of education, the world today has regarded sustainable development as an international consensus, and the sustainability of education has become a common pursuit in the global education field. The process of learning and being educated runs through one's entire life journey. Without educational resources, human survival and reproduction will not be guaranteed.” Has been
- Page 18, line 674-679:“Based on the findings of this study, the research team believes that it is necessary to conduct in-depth research on more samples and explore more potential factors, including personal information security, personalized services, etc. In addition, the design and implementation of sustainable development activities must carefully consider the limitations of fragile sustainable development models and properly address the resulting complexities.” Has been revised.
- Page 2, line 94-96: “Appendix A” Has been revised.

Reviewer 3 Report
Comments and Suggestions for Authors
I suggest to reject the paper as it has nothing to do with the topic of the special issue.
Author Response
Dear Reviewer:
Thank you and the reviewers for your valuable and helpful suggestions concerning our manuscript entitled “A study on the factors influencing the sustainable development of education in the context of COVID-19: Tencent Conference Online Platform” (Article, Manuscript ID: 2947421). We have discussed and studied the reviewers' comments in detail and addressed all the issues raised by the reviewers. The manuscript has been revised accordingly and the changes are highlighted in red. Detailed responses to the reviewers' comments are as follows.
Reviewer: 1
I suggest to reject the paper as it has nothing to do with the topic of the special issue.
[Reply] First of all, I would like to thank you for your careful review of our paper and your feedback. I understand that you think the paper "Factors affecting the sustainable development of education in the context of COVID-19 - Tencent Conference Online Platform" may not fully fit the theme of the special issue "Sustainability in Higher Education: Curriculum Design and Material Development".
However, I believe our research provides insights into the current practical challenges facing higher education, particularly in the context of the global pandemic, which are central to the discussion of educational sustainability. Our paper focuses on one online platform - Tencent Conferences - a tool that has been widely used globally and has played an important role in supporting educational activities during the pandemic.
First, the application of technology in education. This research involves the online platform of Tencent Conference, which is an important part of modern teaching tools and is directly related to the technology integration in course design.
Second, emergency education mode. The COVID-19 pandemic has forced educational institutions to quickly turn to online teaching, which is a case study in how educational materials can be designed and implemented in emergency situations.
Third, accessibility and equity in education. The use of online learning platforms highlights the issue of the digital divide, which is critical to ensuring sustainable and equitable access to educational resources for all students.
Fourth, the innovation and adaptation of curriculum materials. During the pandemic, teachers had to adapt quickly to the new environment, creating and adapting teaching materials to the virtual classroom, which is an example of adaptability and innovation in the development of materials.
Fifth, teaching quality and student participation. How online platforms affect students' learning experiences and outcomes is one of the key factors in assessing educational sustainability.
Sixth, the evaluation and improvement of distance education. Your research may provide insight into the effectiveness of online teaching, which is essential for the continuous improvement of course design and material development in distance education.
Therefore, the study does address the use of technology in the course design process, as well as how to effectively develop and utilize teaching materials in emergency situations.
We recognize that there may be a need to further highlight the links between our research and the topic of the special issue and are willing to make the necessary revisions in accordance with your guidance to ensure that our work is more aligned with the goals of the special issue and the interests of our readers. If possible, we would appreciate your specific suggestions on how to improve our paper to better fit the scope of the special issue.
We sincerely hope to be given the opportunity to reconsider and look forward to your valuable comments.
We have addressed all questions according to the Reviewers’ suggestion and revised the manuscript accordingly. We hope the revised manuscript can be accepted by the reviewers. Once again, thank you very much for your helpful comments and suggestions.
-------------
Yanlong Guo
Social Innovation Design Research Centre,
Anhui University, Hefei 230601,China
Phone: 00-86-152 5655 6306, E-mail: [email protected]
Specification of corrections:
- Page 1, line 11-17:“The difficulties of offline education caused by the COVID-19 epidemic are attracting increasing public attention. Although the international research on online education platforms is gradually deepening, the research on the influencing factors of Chinese users' participation in these platforms shows different results due to regional differences. Therefore, this study adopts the technology acceptance model (TAM) as the basis to build the index system of influencing factors for Tencent conference online education platform. The questionnaire design covered five dimensions, including external environment, platform satisfaction and continuous use intention.” Has been revised
- Page 1, line 22-23:“First, perceived ease of use is the most critical factor influencing users' use of online delivery platforms.” Has been revised
- Page 1, line 32-33:“In the winter of 2019, cases of the novel coronavirus have been reported in Wuhan, Hubei Province, China.” Has been revised
- Page 1, line 34-38:“The outbreak is now having a profound impact on the health, work, and education of people around the world. The United Nations 2030 Agenda for Sustainable Development is an international initiative to address increasingly pressing global challenges and promote sustainable development worldwide.” Has been revised
- Page 2, line 78-88:“Online education platforms independently developed by Chinese universities or jointly initiated by social entities mainly include China University MOOC, Xuetang Online, Good University Online, and Superior Course Alliance. The representative online education platforms developed and operated by Internet enterprises mainly include: NetEase Cloud Class, Tencent Class, New Oriental Online, etc. At the same time, educational platforms in other countries have their own unique advantages. For example, platforms such as Khan Academy and Coursera in the United States are favored by users around the world for their rich course resources and high-quality teaching content. Future Learn in the UK has attracted many users with its flexible learning methods and diverse course content. In contrast, China's education platforms have significant advantages in terms of content richness, teaching methods, and user experience.” Has been revised
- Page 2, line 93-97:“Tencent Conferences feature its high-definition video and audio calling capabilities, allowing users to interact almost without latency. Its flexible screen sharing feature enables teachers to present teaching content in real time. At the same time, with instant messaging and group discussion room functions, students can easily have group discussions or communicate one-on-one with teachers.” Has been revised
- Page 2, line 105-111:“The overall aim of this study was to explore the key factors affecting the sustainable development of education in the context of the COVID-19 pandemic, with a particular focus on the use of online conferencing software. Tencent Conference was chosen as the research object because it was widely adopted as a major tool for teaching and business communication during the epidemic. The purpose of the study was to assess user per-ceptions of the software, as user acceptance and satisfaction directly affect the continued application and effectiveness of online software in the field of education.” Has been revised
- Page 2, line 112-118:“Indeed, if users are found to have a positive attitude towards online conferencing software, this indicates that the software is more likely to be adopted by university faculty and business professionals. This is especially important for higher education institutions, as it involves curriculum design, innovation in teaching methods, and effective use of distance education resources. Furthermore, as educational models continue to evolve, understanding how these tools meet the needs of different user groups is critical to developing sustainable and inclusive pedagogical solutions.” Has been revised
- Page 2, line 120-121:“To date, academics across the globe have highlighted the importance of online teaching in terms of educational sustainability.” Has been revised
- Page 2, line 122-125:“Online teaching broadens the scope and methods of education and learning. [15] Based on digital technology, educators in Europe are upgrading their digital skills through online teaching models.” Has been revised
- Page 3, line 164-168:“Positive progress is being made in integrating the concept of sustainable development into high-quality education and online teaching in schools. At the same time, network lecture is also affected by teaching space, time, content and way. This is reflected in the fact that students often need to interact with multiple teachers and the diversity of online platforms that teachers choose.” Has been revised
- Page 3, line 203-205:“Influenced by the COVID-19 pandemic and the international political and economic environment, global student mobility has shown a new trend.” Has been revised
- Page 4, line 238-241:“Specific methods and processes (Figure 1). Figure 1. Research and analysis framework.” Has been revised
- Page 5, line 272-275:“Analytic hierarchy Process (AHP). [51] AHP is a decision analysis tool designed to deal with complex problems through case-by-case analysis. It combines quantitative and qualitative analysis to determine the weight or priority of individual factors. [52]” Has been revised
- Page 6, line 302-307:“Based on TAM (Technology Acceptance Model) as the basic framework, based on the preliminary analysis of the questionnaire survey results and the in-depth understanding of Tencent conference user experience. The paper constructs a model of influencing factors of Tencent meeting usage behavior. The model covers key factors such as perceived usability, perceived ease of use, external environmental impact, platform satisfaction, and willingness to continue using the platform.” Has been revised
- Page 10, line 405-409:“Therefore, this study adopts the evaluation criteria of analytic hierarchy process as a method to determine the relative position and distribution of project elements in the platform hierarchy model. With this approach, we were able to quantify research results and assess the importance of design elements. The research team used mathematical tools to construct the data into matrices for processing.” Has been revised
- Page 10, line 430-433:“These results are crucial for determining the importance ranking of factors in design decisions. Inconsistencies may occur when comparing the importance of different factors. In order to ensure the reliability of the calculated results, we performed a consistency check on the calculated data.” Has been revised
- Page 16, line 564-567:“Focusing on Tencent conference software, this paper investigates and analyzes the user behavior and preferences of online lecture platform in the context of sustainable education. Based on this research background, the following conclusions are drawn.” Has been revised
- Page 17, line 598-602:“Teachers should actively adopt the concept of sustainable development and become the key force in promoting sustainable education. At the same time, as the successors of the future, young students shoulder the primary responsibility for promoting sustainable development.” Has been revised
- Page 17, line 607-612:“Many participants lack experience in online teaching, which presents significant challenges to this emerging teaching model. To realize the sustainable development of vocational education, we must design both practical and efficient talent training programs. Online teaching platform because of its ability to provide instant, rapid feedback and other advantages, so that users tend to give a more positive evaluation.” Has been revised
- Page 17, line 620-628:“Fifth, about the interviewees' personal evaluation. The rapid growth of educational online platforms and the key role they play in respondents' lives. As an educator, I was initially apprehensive about the sudden change, but soon became attracted to the potential and flexibility that online education offered. Virtual classrooms can transcend geographical limitations and provide students with resources and lecturers from around the world. However, technical issues, student engagement, and differences in home environments also pose challenges. Respondents' overall assessments of the experience were mixed, ranging from frustration with current limitations to optimism about future possibilities.” Has been revised
- Page 17, line 637-640:“Educational resources are the foundation of sustainable development of education, and human resources are its core. The whole process of education should always take humanism as the core, and its ultimate goal is to cultivate capable talents.” Has been revised
- Page 17-18, line 641-673:“Network teaching not only provides people with a convenient way of communication, but also releases the huge space of its development potential. Considering the sustainability of education, the world today has regarded sustainable development as an international consensus, and the sustainability of education has become a common pursuit in the global education field. The process of learning and being educated runs through one's entire life journey. Without educational resources, human survival and reproduction will not be guaranteed.” Has been revised
- Page 18, line 674-679:“Based on the findings of this study, the research team believes that it is necessary to conduct in-depth research on more samples and explore more potential factors, including personal information security, personalized services, etc. In addition, the design and implementation of sustainable development activities must carefully consider the limitations of fragile sustainable development models and properly address the resulting complexities.” Has been revised
- Page 2, line 94-96: “Appendix A” Has been revised
Round 2
Reviewer 2 Report
Comments and Suggestions for Authors
Thanks for revising the manuscript taking into account the suggestions provided in the first round.
Comments on the Quality of English Languagen/a
Author Response
Dear Reviewer:
Thank you and the reviewers for your valuable and helpful suggestions concerning our manuscript entitled “A study on the factors influencing the sustainable development of education in the context of COVID-19: Tencent Conference Online Platform” (Article, Manuscript ID: 2947421). We have discussed and studied the reviewers' comments in detail and addressed all the issues raised by the reviewers. The manuscript has been revised accordingly and the changes are highlighted in red. Detailed responses to the reviewers' comments are as follows.
Reviewer: 1
Thanks for revising the manuscript taking into account the suggestions provided in the first round.
[Reply] I really appreciate your feedback and your attention to the changes I made to the manuscript. I am glad to hear that these changes have been approved by you and have helped to improve the quality of the paper.
Please know that I welcome and appreciate your valuable suggestions during the judging process. This feedback not only provided me with the opportunity to improve my work, but also ensured that my research better matched the topic and quality requirements of the special issue.
If you have any further suggestions or questions, please feel free to let us know. I am looking forward to the next step of the manuscript and hope to get the opportunity to publish it.
Thank you again for your time and effort.
We have addressed all questions according to the Reviewers’ suggestion and revised the manuscript accordingly. We hope the revised manuscript can be accepted by the reviewers. Once again, thank you very much for your helpful comments and suggestions.
-------------
Yanlong Guo
Social Innovation Design Research Centre,
Anhui University, Hefei 230601,China
Phone: 00-86-152 5655 6306, E-mail: [email protected]
Specification of corrections:
- Page 2, line 77-82:“The team's findings underscore the importance of interdisciplinary collaboration in developing and supporting online learning platforms, which echoes the special issue's emphasis on interdisciplinary education. In addition, the findings provide valuable in-sights for education policymakers and curriculum designers who can use them to improve learning outcomes when designing and implementing sustainable distance education solutions.” Has been
- Page 2, line 87-90:“This research not only provides a theoretical and empirical basis for understanding and promoting the sustainable use of online education platforms, but also provides practical recommendations for the sustainable development of higher education, thereby directly supporting the core objectives and scope of this special issue.” Has been
- Page 18, line 540-548:“Through the empirical analysis of Tencent Conference online education platform, this study directly responds to the concern of this special issue on the sustainability of higher education. In the context of the disruption of traditional education models caused by COVID-19, this study reveals how online education platforms can become a key component of education for sustainable development. Through the Technology Acceptance Model (TAM), the research team not only identified the key factors that influence the willingness to continue using online education platforms, but also provided practical strategies for higher education institutions to promote the effectiveness and sustainability of distance learning.” Has been
Reviewer 3 Report
Comments and Suggestions for Authors
After carefully reviewing the paper, I regret to inform you that it does not align with the theme of the special issue. The focus on sustaining education during COVID-19 does not relate to the core objective of the special issue, which centers around the development and implementation of sustainability competencies in education.
Author Response
Dear Reviewer:
Thank you and the reviewers for your valuable and helpful suggestions concerning our manuscript entitled “A study on the factors influencing the sustainable development of education in the context of COVID-19: Tencent Conference Online Platform” (Article, Manuscript ID: 2947421). We have discussed and studied the reviewers' comments in detail and addressed all the issues raised by the reviewers. The manuscript has been revised accordingly and the changes are highlighted in red. Detailed responses to the reviewers' comments are as follows.
Reviewer: 1
After carefully reviewing the paper, I regret to inform you that it does not align with the theme of the special issue. The focus on sustaining education during COVID-19 does not relate to the core objective of the special issue, which centers around the development and implementation of sustainability competencies in education.
[Reply] I am very grateful for your careful review and valuable comments on my manuscript. I understand your question about the consistency of the article with the theme of the special issue, and I respect your point of view. I will add a paragraph to emphasize how this article contributes to the scope of this special issue.
I would like to clarify that my research aims to explore the role and impact of online education platforms on educational sustainability in the context of COVID-19. By applying a technology acceptance model (TAM) to construct an index system of influencing factors for Tencent Conference online education platforms, this study provides insight into the willingness to continue using online education platforms, which is critical to understanding and promoting sustainability in higher education
I believe that, especially in the context of the current global health crisis, the continuity and adaptability of education faces unprecedented challenges. Therefore, my thesis aims to provide strategies and insights for higher education institutions to better utilize online platforms to maintain the continuity of educational activities. I believe these findings have important implications for understanding and supporting sustainable capacity development in education.
I am willing to further revise my paper based on feedback from you and the editorial team to ensure that it better meets the goals of the special issue. I would be grateful if you could provide specific advice or guidance.
Thank you again for your time and consideration.
Reviewer: 2
Academic Editors, who suggested adding an additional paragraph emphasizing on
how this paper contributes to the scope of this Special Issue.
[Reply] Thanks to the suggestions of the reviewers, the research team will add a paragraph to the paper that will clearly set out how the research contributes to the topic of the special issue. The changes are as follows: “The team's findings underscore the importance of interdisciplinary collaboration in developing and supporting online learning platforms, which echoes the special issue's emphasis on interdisciplinary education. In addition, the findings provide valuable insights for education policymakers and curriculum designers who can use them to improve learning outcomes when designing and implementing sustainable distance education solutions.”
The changes are as follows: “This research not only provides a theoretical and empirical basis for understanding and promoting the sustainable use of online education platforms, but also provides practical recommendations for the sustainable development of higher education, thereby directly supporting the core objectives and scope of this special issue.”
The changes are as follows: “Through the empirical analysis of Tencent Conference online education platform, this study directly responds to the concern of this special issue on the sustainability of higher education. In the context of the disruption of traditional education models caused by COVID-19, this study reveals how online education platforms can become a key component of education for sustainable development. Through the Technology Acceptance Model (TAM), the research team not only identified the key factors that influence the willingness to continue using online education platforms, but also provided practical strategies for higher education institutions to promote the effectiveness and sustainability of distance learning.”
We have addressed all questions according to the Reviewers’ suggestion and revised the manuscript accordingly. We hope the revised manuscript can be accepted by the reviewers. Once again, thank you very much for your helpful comments and suggestions.
-------------
Yanlong Guo
Social Innovation Design Research Centre,
Anhui University, Hefei 230601,China
Phone: 00-86-152 5655 6306, E-mail: [email protected]
Specification of corrections:
- Page 2, line 77-82:“The team's findings underscore the importance of interdisciplinary collaboration in developing and supporting online learning platforms, which echoes the special issue's emphasis on interdisciplinary education. In addition, the findings provide valuable in-sights for education policymakers and curriculum designers who can use them to improve learning outcomes when designing and implementing sustainable distance education solutions.” Has been
- Page 2, line 87-90:“This research not only provides a theoretical and empirical basis for understanding and promoting the sustainable use of online education platforms, but also provides practical recommendations for the sustainable development of higher education, thereby directly supporting the core objectives and scope of this special issue.” Has been
- Page 18, line 540-548:“Through the empirical analysis of Tencent Conference online education platform, this study directly responds to the concern of this special issue on the sustainability of higher education. In the context of the disruption of traditional education models caused by COVID-19, this study reveals how online education platforms can become a key component of education for sustainable development. Through the Technology Acceptance Model (TAM), the research team not only identified the key factors that influence the willingness to continue using online education platforms, but also provided practical strategies for higher education institutions to promote the effectiveness and sustainability of distance learning.” Has been